# Waveform Design for Space–Time Coded MIMO Systems with High Secrecy Protection †

**Pingping Shang, Hyein Lee** 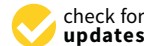 **and Sooyoung Kim *** 

IT Convergence Research Center, Division of Electronics Engineering, Jeonbuk National University, 567 Baekje-daero, Deokjin-gu, Jeonju 54896, Jeollabuk-do, Korea; pingajiayou@jbnu.ac.kr (P.S.); leehyein96@jbnu.ac.kr (H.L.)
* Correspondence: sookim@jbnu.ac.kr
† This paper is an extended version of our paper published in ICTC 2020.

**Abstract:** In this paper, we present a new secrecy-enhancing scheme for multi-input-multi-output (MIMO) systems using a space–time coding scheme. We adopt a quasi-orthogonal space–time block coding (QO-STBC) scheme that was originally designed to improve the performance of the MIMO system, and propose an efficient waveform design that can enhance the secrecy, as well as improve the error rate performance. Channel- and signal-dependent artificial interference (AI) is added to the proposed waveform, so that only a legitimate receiver can successfully retrieve information. We investigate the secrecy capacity of the proposed scheme, and demonstrate that the proposed scheme provides highly enhanced secrecy performance, compared to the conventional schemes. The performance simulation results reveal that the transmitted information can be properly extracted only at the legitimate receiver.

**Keywords:** multi-input-multi-output (MIMO); space time block coding; physical layer security (PLS); secrecy capacity

## 1. Introduction

Most technology development in wireless systems has mainly focused on the enhancement of spectral efficiency and/or power efficiency (error rate performance). The explosion of multiple-input multiple-output (MIMO) technology can be one of the representative examples that have largely contributed to enhancing spectral or power efficiency. Due to their inherent broadcasting nature, wireless communication systems are vulnerable to security and privacy protection. However, waveform design for the past generation systems has not focused mainly on the secrecy performance aspect. Although the classical cryptography protocols in the network layer may guarantee secure communications, the involved secret-key distribution and management processes are generally unaffordable, and are fragile to attacks, especially in wireless systems [1]. For this reason, security protection needs to be considered during the waveform design process along with power and spectral efficiency.

Physical layer security (PLS) schemes can be considered as one of the effective waveform design techniques to provide security protection, and a number of study results have been reported by exploiting the characteristics of wireless channels [2–4]. Because MIMO techniques are now almost mandatory in most wireless systems, PLS schemes combined with MIMO techniques have recently gained a lot of attention. As a measure of secrecy protection, Reference [5] derived the secrecy capacity of a MIMO system. Later, Reference [6] proved that perfect secrecy protection could theoretically be achieved for MIMO systems, and these results accelerated the development of PLS techniques for MIMO systems. Reference [7] considered utilization of spatial modulation (SM) and adding artificial

noise (AN) as effective means to increase the secrecy capacity for MIMO systems. The essence of AN technology is to add a noise that is orthogonal to the legitimate channel, so that it can be null to the legitimate receiver, while presenting interference to the illegal receiver.

On the other hand, there have been efforts to utilize space–time block coding (STBC) schemes for PLS [8–13]. The Alamouti scheme was utilized for PLS [8–12]. As a cooperative multi-user transmission technique, Reference [8] proposed the idea of transmit antenna selection (TAS)-Alamouti scheme for PLS enhancement. Reference [9] proposed another cooperative Alamouti scheme, where Alamouti users collaborate with each other with an AN-aided technique to secure the transmissions. They demonstrated that the secrecy sum-capacity was improved by adding AN aligned with the null space of the legitimate channel matrix. The Alamouti scheme was also utilized in SM with AN, where SM was used to increase the secrecy performance while the Alamouti scheme was used to enhance the performance [10]. Reference [11] proposed a PLS scheme employing a coordinate interleaved orthogonal design, while Reference [12] proposed a transmission technique for PLS by applying the Alamouti coded non-orthogonal multiple access scheme.

Furthermore, secrecy outage probability was derived for a quasi-orthogonal STBC (QO-STBC) scheme that linearly combines two Alamouti codes [13]. The results in this research demonstrate the secrecy outage probability according to the power ratio of legitimate and illegal receivers. This QO-STBC scheme utilized a power scaling (PS) method, but the PS itself did not contribute to enhancing the secrecy performance. Instead, the PS was used to achieve the full rate and full diversity effect. Reference [14] also proposed a similar idea, presenting a new encoding matrix for QO-STBC to achieve full orthogonality of the channel matrix.

In this paper, we propose a new waveform design for the MIMO system using STBC, which is targeted at enhancing the secrecy, as well as the error rate performance. We modify the idea of combing AN with STBC, and adopt a QO-STBC scheme with linear detection capability [14]. The advantage of the QO-STBC scheme in [14] lies in the full orthogonality of the channel matrix, and thus it was termed a linear decoding QO-STBC (LD-QO-STBC) scheme. However, its disadvantage lies in the non-uniform signal power distribution. In the proposed scheme, we utilize this property, and add the artificial interference (AI) by considering non-uniform signal distribution. The waveform with time-varying AI is received at the legitimate receiver without any interference, while the added interference at the illegal receiver prevents proper detection.

We note that STBC schemes are widely used in modern wireless systems to improve performance. Even though the STBC schemes were investigated for PLS in the previous studies, they can achieve reasonable secrecy protection only with a limited condition, i.e., when the power strength of the legitimate receiver is sufficiently higher than that of the illegal receiver. An approach with AN can be used, but this requires additional power allocation. Therefore, the proposed PLS scheme can be efficiently utilized for many wireless systems equipped with STBC, without any performance degradation.

Compared to the conventional PLS schemes with STBC, we summarize the superiority of our work as follows. First, the proposed scheme can be utilized in the STBC MIMO systems with more than two transmit antennae, and thus it can achieve more diversity gain compared to the previous works with the Alamouti scheme [8–12]. Second, the proposed scheme does not allocate additional power to AN for security protection, and thus it is power efficient. The probability distribution of the LD-QO-STBC signal waveform is newly derived. By using the investigation results of the signal distribution, the power reallocation is dynamically performed across the multi transmit antennae. This power reallocation contributes to equalizing power distribution across the transmit antennae as well as transmit time. In addition, it imposes severe interference on the eavesdropper. The novelty of the proposed idea lies in using the new waveform for STBC MIMO systems with power reallocation. By virtue of the above two strong advantages, the proposed scheme can achieve perfect security protection even though the eavesdropper has almost the same power as the legitimate receiver.

The rest of this paper is organized as follows. Section 2 describes the related works with the proposed scheme. We first introduce the PLS scheme that combines the Alamouti-code with AN, and present the LD-QO-STBC scheme [14]. Section 3 presents the proposed waveform design for PLS using the LD-QO-STBC scheme. We first investigate the characteristics of the signal constellation of the LD-QO-STBC scheme. Afterwards, we propose a method to generate signal- and channel-dependent AI that can be perfectly canceled only at the legitimate receiver. Section 4 evaluates the secrecy capacity of the proposed scheme, and compares it with that of the conventional scheme. In addition, we present a bit error rate (BER) performance simulation results to demonstrate proper security protection. Finally, Section 5 draws the conclusions.

Notation: Bold lower case letters represent vectors, while bold upper case letters denote matrices. $\mathbf{A}^T$, $\mathbf{A}^H$, and $\|\mathbf{A}\|$ denote the transpose, Hermitian transpose, and Euclidean norm of matrix $\mathbf{A}$, respectively. The superscript $(\cdot)^*$ denotes the complex conjugate, while $*$ denotes the convolution operator.

## 2. Related Works

### 2.1. Alamouti Coded PLS with AN

The generic model for the PLS scheme, comprises a cooperative wireless network that consists of three nodes. Suppose that a legitimate transmitter node is referred to as Alice, which is the source node. The corresponding destination node is referred to as Bob, which is the legitimate receiver node. On the other hand, the third node, named Eve, is the passive eavesdropper node.

Figure 1 shows a block diagram for the Alamouti-coded PLS with AN [10]. In this system, it is assumed that Alice is equipped with three antennae, while Bob and Eve are equipped with only one antenna, respectively. At the transmitter, the bit stream emitted by a binary source is divided into a block containing one bit to select one antenna among Ant 1 and Ant 2 for SM, and the other blocks containing information bits to transmit. The selected antenna will form the first virtual antenna for the Alamouti scheme, while Ant 3 in Figure 1 will form the second virtual antenna for the $2 \times 1$ Alamouti scheme. Therefore, the first virtual antenna is dynamically changed by time. During the first symbol period, $s_1$ and $s_2$ which are mapped by using $m$ bits for each will be transmitted through two antennae, respectively. In addition, AN will be injected for secrecy purpose. During the second symbol period, the Alamouti encoded symbols, $-s_2^*$ and $s_1^*$, are transmitted along with ANs through two antennae, respectively.

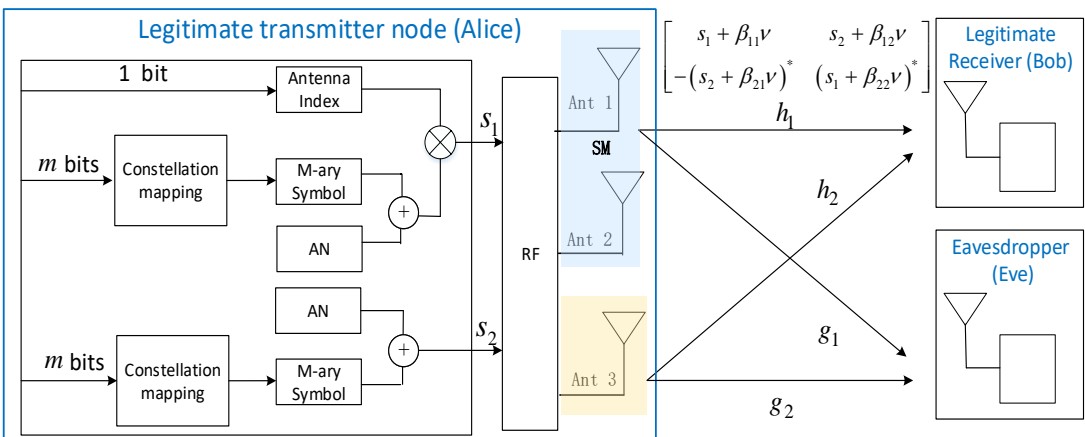

**Figure 1.** System model of the Alamouti-coded physical layer security (PLS) with artificial noise (AN).

It is assumed that the channel coefficients are constant within each transmission block, i.e., two symbol periods, and they are changed independently from block to block, as in the conventional

Alamouti scheme. The perfect channel state information (CSI) is assumed to be available at Bob and Eve, but it may not be available at Alice. The encoding matrix for the Alamouti code with AN, $\mathbf{X}_{AN}$ can be expressed as:

$$\mathbf{X}_{AN} = \begin{bmatrix} s_1 + \beta_{11}v & s_2 + \beta_{12}v \\ -(s_2 + \beta_{21}v)^* & (s_1 + \beta_{22}v)^* \end{bmatrix}, \tag{1}$$

where $s_1$ and $s_2$ are modulation symbols used for the Alamouti code during two symbol periods, and they are drawn from the *M*-ary quadrature amplitude modulation (QAM) constellation, where $M = 2^m$. In addition, $v$ is the complex Gaussian AN with zero-mean and unit variance, and $\beta_{ij}$s are the coefficients of the $v$ for the *i*-th timeslot at the *j*-th antenna. The coefficients $\beta_{ij}$s are designed to be nulled at the legitimate receiver. Therefore they should be computed by exploiting the CSI of the legitimate channel, which results in the following two possible sets of solutions for each symbol period, i.e., for the first time period:

$$\beta_{11} = h_2, \ \beta_{12} = -h_1, \ \text{or} \ \beta_{11} = -h_2, \ \beta_{12} = h_1, \tag{2}$$

and for the second time period:

$$\beta_{21} = h_2^*, \ \beta_{22} = h_1^*, \ \text{or} \ \beta_{21} = -h_2^*, \ \beta_{22} = -h_1^*, \tag{3}$$

where, $h_1$ and $h_2$ are the channel gains from the first and second virtual antennae at Alice to Bob, respectively.

With the above solution, the received signal at Bob during two symbol periods, $y_{b1}$ and $y_{b2}$ can be represented by:

$$\begin{aligned} y_{b1} &= h_1 s_1 + h_2 s_2 + n_{b1}, \\ y_{b2} &= -h_1 s_2^* + h_2 s_1^* + n_{b2}, \end{aligned} \tag{4}$$

where $n_{b1}$ and $n_{b2}$ are additive white Gaussian noise (AWGN) at Bob for the first and second symbol periods, respectively. Because the added ANs are perfectly canceled due to (2) and (3) at Bob, the detection can be successfully made as in the conventional $2 \times 1$ Alamouti scheme. On the other hand, the received signal at Eve will contain serious interference terms as follows:

$$\begin{aligned} y_{e1} &= g_1 (s_1 + \beta_{11}v) + g_2 (s_2 + \beta_{12}v) + n_{e1}, \\ y_{e2} &= -g_1 (s_2 + \beta_{21}v)^* + g_2 (s_1 + \beta_{22}v)^* + n_{e2}, \end{aligned} \tag{5}$$

where $g_1$ and $g_2$ are the channel gains from the first and second virtual antennae at Alice to Eve, respectively. In addition, $n_{e1}$ and $n_{e2}$ are AWGN at Eve for the first and second symbol periods, respectively.

### 2.2. Linear Decoding QO-STBC Scheme

After Alamouti first proposed the orthogonal STBC scheme with two transmit antennae [15], efforts were made to invent full-rate STBC schemes with a larger number of transmitting antennae. In an effort to increase the antenna size and achieve the full data rate, by loosening the orthogonality condition, several QO-STBC schemes have been proposed [16,17]. These QO-STBC schemes could achieve the full rate, but at the cost of higher decoding complexity, due to the non-perfect orthogonality of the encoding matrix.

In order to solve this problem, a new QO-STBC scheme providing a simple linear detection capability was proposed [14]. By applying the Givens rotation to the detection matrix of a conventional

QO-STBC scheme, new encoding matrices for three and four antenna systems were derived. The encoding matrix, **X**, for four transmit antennae can be expressed as follows:

$$\mathbf{X} = \begin{bmatrix} s_1 + s_3 & s_2 + s_4 & s_3 - s_1 & s_4 - s_2 \\ -s_2^* - s_4^* & s_1^* + s_3^* & s_2^* - s_4^* & s_3^* - s_1^* \\ s_3 - s_1 & s_4 - s_2 & s_1 + s_3 & s_2 + s_4 \\ s_2^* - s_4^* & s_3^* - s_1^* & -s_2^* - s_4^* & s_1^* + s_3^* \end{bmatrix}. \tag{6}$$

As shown above, a signal transmitted for each time slot at each antenna has now become a linear combination of two modulation symbols. By assuming a single receive antenna, the received signal during four time slots can be expressed as

$$[y_1\ y_2\ y_3\ y_4]^T = \mathbf{X}\,[h_1\ h_2\ h_3\ h_4]^T + \mathbf{n}^T, \tag{7}$$

where $h_i$ is the channel gain from the $i$-th transmit antenna to the receive antenna, and $\mathbf{n} = [n_1\ n_2\ n_3\ n_4]$ is an AWGN vector for the four time slots. For detection, the received signal vector can be represented by using the equivalent channel matrix, **H**, as follows:

$$[y_1\ y_2^*\ y_3\ y_4^*]^T = \mathbf{H}\,[s_1\ s_2\ s_3\ s_4]^T + [n_1\ n_2^*\ n_3\ n_4^*]^T, \tag{8}$$

where

$$\mathbf{H} = \begin{bmatrix} h_1 - h_3 & h_2 - h_4 & h_1 + h_3 & h_2 + h_4 \\ h_2^* - h_4^* & h_3^* - h_1^* & h_2^* + h_4^* & -h_1^* - h_3^* \\ h_3 - h_1 & h_4 - h_2 & h_1 + h_3 & h_2 + h_4 \\ h_4^* - h_2^* & h_1^* - h_3^* & h_2^* + h_4^* & -h_1^* - h_3^* \end{bmatrix}. \tag{9}$$

The channel matrix **H** is orthogonal, although the encoding matrix **X** is quasi-orthogonal. For this reason, maximum likelihood (ML) decoding can be achieved via simple linear detection, as given by:

$$\hat{\mathbf{s}} = \mathbf{H}^H\,[y_1\ y_2^*\ y_3\ y_4^*]^T. \tag{10}$$

## 3. Secrecy-Enhancing LD-QO-STBC Scheme

### 3.1. Waveform Analysis for LD-QO-STBC

In order to design an efficient waveform for secrecy enhancement, we first investigate the signal constellation of the LD-QO-STBC scheme. By letting combinations of modulation symbols be used in (6) as follows:

$$\begin{aligned} x_1 &= s_1 + s_3, \quad x_2 = s_2 + s_4, \\ x_3 &= s_3 - s_1, \quad x_4 = s_4 - s_2, \end{aligned} \tag{11}$$

the encoding matrix **X** in (6) can be represented as:

$$\mathbf{X} = \begin{bmatrix} x_1 & x_2 & x_3 & x_4 \\ -x_2^* & x_1^* & -x_4^* & x_3^* \\ x_3 & x_4 & x_1 & x_2 \\ -x_4^* & x_3^* & -x_2^* & x_1^* \end{bmatrix}. \tag{12}$$

Because each element of **X** is a linear combination of any possible values of $s_i$ and $s_j$, the waveforms of $s_i$ and $x_i$ will be different. In order to investigate the signal distributions, we denote constellation symbol sets used for $s_i$ and $x_i$ as $a \in O_a$ and $b \in O_b$, respectively. We note that for any Gray-coded QAM constellation symbol mapping, $s_i$, $-s_i$, $s_i^*$, and $-s_i^*$ are all in the same set $O_a$, due to its symmetric constellation. Likewise, elements of **X** that are either $x_i$, $-x_i$, $x_i^*$, or $-x_i^*$, are in the same set $O_b$. Nevertheless, it is clear that $O_b$ forms a different set from $O_a$.

Figure 2 demonstrates an example of waveform transformation from $O_a$ to $O_b$ when 4-QAM is used for $a$. In the figure, we denote $a_i$ and $b_i$ as one of the complex constellation symbols included in $O_a$ and $O_b$, respectively. In addition, we derive the probability density function (PDF) of $b$, and show that it is not a uniform distribution. For any $M$-ary QAM, we can assume that without loss of generality, the PDF of $a$ follows a uniform distribution. That is,

$$f_a(a) = 1/M, \quad a \in O_a. \tag{13}$$

By assuming the independency of the real and imaginary parts of $a$,

$$f_{a^R}(a^R) = 1/\sqrt{M}, \quad f_{a^I}(a^I) = 1/\sqrt{M}, \quad a \in O_a, \tag{14}$$

where $a^R$ and $a^I$ are the real and imaginary parts of a complex value $a$, respectively.

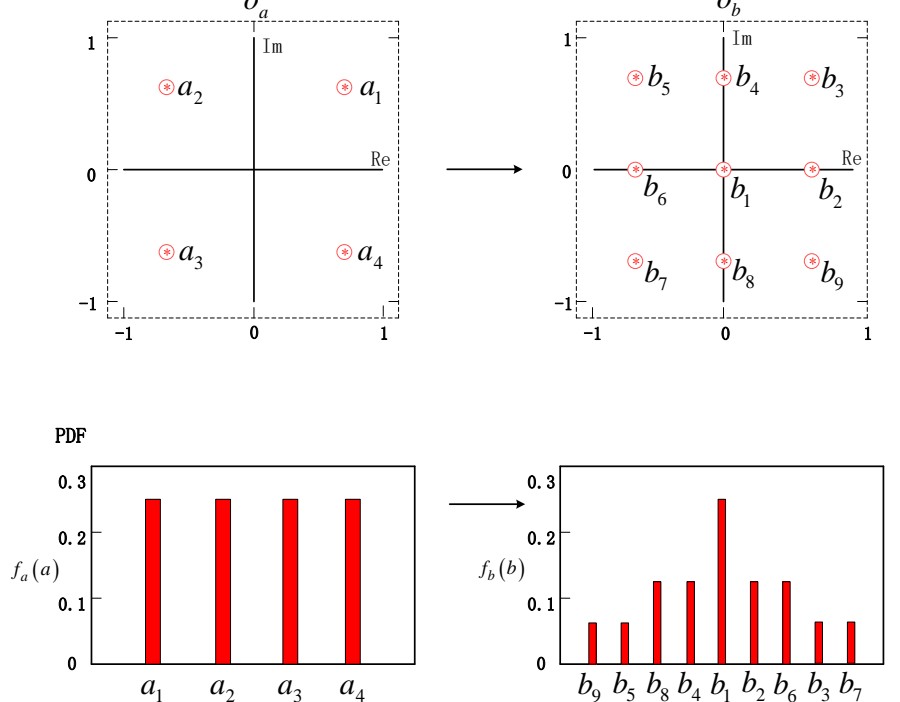

**Figure 2.** Signal constellations and PDFs for $a \in O_a$ and $b \in O_b$ used for $s_i$ and $x_i$, respectively.

For any symbol $b$, we note that it is the sum of two independent random variables, so its PDF can be found by first taking the convolution of $f_{b^R}(b^R)$ and $f_{b^I}(b^I)$ in their real and imaginary parts, respectively. Therefore,

$$\begin{aligned} f_{b^R}(b^R) &= f_{a^R}(a^R) * f_{a^R}(a^R), \\ f_{b^I}(b^I) &= f_{a^I}(a^I) * f_{a^I}(a^I), \quad a \in O_a, \end{aligned} \tag{15}$$

and finally using the independency of real and imaginary parts,

$$f_b(b) = f_{b^R}(b^R)f_{b^I}(b^I), \quad b \in O_b. \tag{16}$$

Figure 2 compares the PDFs, $f_a(a)$ and $f_b(b)$ when 4-QAM is used for $O_a$. We find that the number of elements in $O_b$ is greater than that in $O_a$, and that there is a null value $0 + j0$ in $O_b$, which is denoted by $b_1$ in the example of 4-QAM of Figure 2. Whatever $M$-ary QAM is used for $O_a$, this null value exists. We also find that the probability of having the null value of $b$ is the maximum among the other possible values, i.e., $f_b(b_1)$ is the maximum amongst, and Figure 2 also demonstrates this. These findings will

be applicable to higher-order *M*-ary QAM. The resultant $O_b$ from the LD-QO-STBC scheme will incur zero-crossing amongst non-zero modulation symbol values.

### 3.2. The Design of Artificial Interference

From the investigation in the previous section, we note that the probability of having the null value of $x_i$, i.e., $x_i = 0 + j0$ is the maximum amongst all possible values in $O_b$. We also note from (6) that whenever we have zero value of $x_i$, there will be $2s_i$ value in the same row in **X**. For example, consider the case when $x_1 = 0$, which means $s_1 = -s_3$, because $x_1 = s_1 + s_3$ as defined in (11). This implies that $x_3 = 2s_3$, because $x_3 = s_3 - s_1$, as in (11). We refer to these two elements that have the null value and $2s_i$ as a pair. Investigation of (6) reveals that there are two pairs in each row; that is, the first and third elements form one pair, and the second and fourth elements form the other pair.

We design the AI in order to satisfy the following condition. First, by adding AI, we escape the zero value transmission. For this, whenever we have a zero value of $x_i$, we add AI that will change the transmitted symbol to a non-zero value. At the same time, we subtract the added value from the paired elements, which reduces the original amplitude. Second, the AI is added in such a way that the paired elements contribute to canceling each other's AI at the receiving end. For this, we multiply the channel gains in such a way that the added AI is canceled.

To achieve the above design goal, we represent the received signal at Bob during the *i*-th time period, $y_{bi}$, as follows:

$$y_{bi} = \sum_{j=1}^{4} \left( \chi_{ij} + \omega_{ij} \right) h_j + n_{bi}, \tag{17}$$

where, $\chi_{ij}$ is the element of **X** in the *i*-th row and *j*-th column, $\omega_{ij}$ is the AI term, and $n_{bi}$ is the noise at Bob during the *i*-th time period. Because $\chi_{ij}$ and $\chi_{i(j+2)}$, $j = 1, 2$, always form a pair, we represent (17) as the following pairwise form:

$$y_{bi} = \sum_{j=1}^{2} \left\{ \left( \chi_{ij} + \omega_{ij} \right) h_j + \left( \chi_{i(j+2)} + \omega_{i(j+2)} \right) h_{j+2} \right\} + n_{bi}. \tag{18}$$

In order to satisfy the first condition, when $\chi_{ij} = 0$, we set $\omega_{ij}$ to a non-zero value, so that it can increase the power of $\chi_{ij}$. At the same time, we decrease the same amount of power from the paired element. In other words, we set $\omega_{ij} = \omega_{i(j+2)} = \chi_{i(j+2)}/2$, in the case $\chi_{ij} = 0$. This pairwise power reallocation will contribute to decreasing the difference of power allocations across the transmit antennae. Even though AI is added, it is a kind of reallocation of power, and thus there will be no additional average power allocation. In addition, thanks to the following solution to the second condition, this AI will be perfectly eliminated.

The second condition requires that AI should be canceled at the legitimate receiver. For this, we represent (18) by separating the signal and AI terms, that is

$$y_{bi} = \sum_{j=1}^{2} \left( \chi_{ij} h_j + \chi_{i(j+2)} h_{j+2} \right) + \sum_{j=1}^{2} \left( \omega_{ij} h_j + \omega_{i(j+2)} h_{j+2} \right) + n_{bi}. \tag{19}$$

Because the AI term in (19) is $\sum_{j=1}^{2} \left( \omega_{ij} h_j + \omega_{i(j+2)} h_{j+2} \right)$, we need to make it zero, i.e.,

$$\sum_{j=1}^{2} \left( \omega_{ij} h_j + \omega_{i(j+2)} h_{j+2} \right) = 0. \tag{20}$$

A simple solution to (20) can be letting $\omega_{ij}$ and $\omega_{i(j+2)}$ contain $-h_{(j+2)}$ and $h_j$ terms, respectively.

By combining the above two conditions at the same time, $\omega_{ij}$ can be defined as follows:

$$
\begin{aligned}
\omega_{ij} &= \frac{\chi_{i(j+2)}h_{j+2}}{2}, \quad \omega_{i(j+2)} = -\frac{\chi_{i(j+2)}h_j}{2}, \quad \text{if } j = 1, 2, \text{and } \chi_{ij} = 0, \\
\omega_{ij} &= \frac{\chi_{i(j-2)}h_{j-2}}{2}, \quad \omega_{i(j-2)} = -\frac{\chi_{i(j-2)}h_j}{2}, \quad \text{if } j = 3, 4, \text{and } \chi_{ij} = 0, \\
\omega_{ij} &= 0, \quad\quad\quad\quad\quad\quad\quad\quad\quad\quad\quad\quad \chi_{ij} \neq 0, \chi_{i(j\pm2)} \neq 0.
\end{aligned}
\tag{21}
$$

As shown above, $\omega_{ij}$ has a non-static time-varying value that is dependent on $\chi_{ij}$, as well as on $h_j$. At the transmitter part, this AI contributes to reducing the difference of power allocations, while at the receiving end, it will be perfectly canceled out. Therefore, by using (21), (18) can be represented without any AI, as follows:

$$
y_{bi} = \sum_{j=1}^{2} \left( \chi_{ij}h_j + \chi_{i(j+2)}h_{j+2} \right) + n_{bi}.
\tag{22}
$$

In summary, the received signal at Bob during the four time slots can be represented as:

$$
[y_{b1}\, y_{b2}\, y_{b3}\, y_{b4}]^T = (\mathbf{X} + \mathbf{W})\, [h_1\, h_2\, h_3\, h_4]^T + \mathbf{n}_b^T,
\tag{23}
$$

where $\mathbf{W}$ is the AI matrix with $\omega_{ij}$ as the element in the $i$-th row and $j$-th column. Then, the received signal after the interference cancelation can be equivalently represented as:

$$
[y_{b1}\, y_{b2}^*\, y_{b3}\, y_{b4}^*]^T = \mathbf{H}\, [s_1\, s_2\, s_3\, s_4]^T + [n_{b1}\, n_{b2}^*\, n_{b3}\, n_{b4}^*]^T,
\tag{24}
$$

and the detection can be made exactly the same as in the conventional detection using (10).

On the other hand, the received signal vector at Eve is expressed as follows:

$$
[y_{e1}\, y_{e2}\, y_{e3}\, y_{e4}]^T = (\mathbf{X} + \mathbf{W})\, [g_1\, g_2\, g_3\, g_4]^T + \mathbf{n}_e^T,
\tag{25}
$$

where $g_j$ is the channel gain from the $j$-th transmit antenna from Alice to Eve. Because the added AI matrix $\mathbf{W}$ has factors related to the channel gains of $h_j$s for Bob, it cannot be canceled out at Eve. That is, the received signal at Eve during the $i$-th time slot is expressed as follows:

$$
y_{ei} = \begin{cases}
\sum_{j=1}^{2} \left\{ \left( \chi_{ij} + \frac{\chi_{i(j+2)}h_{j+2}}{2} \right) g_j + \left( \chi_{i(j+2)} - \frac{\chi_{i(j+2)}h_j}{2} \right) g_{j+2} \right\} + n_{ei}, & \text{if } \chi_{ij} = 0, \\
\sum_{j=1}^{2} \left\{ \left( \chi_{ij} - \frac{\chi_{ij}h_{j+2}}{2} \right) g_j + \left( \chi_{i(j+2)} + \frac{\chi_{ij}h_j}{2} \right) g_{j+2} \right\} + n_{ei}, & \text{if } \chi_{i(j+2)} = 0, \\
\sum_{j=1}^{2} \left\{ \chi_{ij}g_j + \chi_{i(j+2)}g_{j+2} \right\} + n_{ei}, & \text{if } \chi_{ij} \neq 0, \chi_{i(j+2)} \neq 0.
\end{cases}
\tag{26}
$$

Even though, there is the probability of not having AI, i.e., $\chi_{ij} \neq 0$ and $\chi_{i(j+2)} \neq 0$, and thus $\omega_{ij} = 0$, frequent uncanceled AI will cause serious performance degradation.

## 4. Performance Evaluation

This section presents the secrecy and BER performances of the proposed waveform design using LD-QO-STBC over a Rayleigh-faded MIMO channel, and compares them with the conventional schemes. In order to consider the worst-case for secrecy protection, we assume that not only Bob but also Eve have perfect CSI, and that they have the same signal-to-noise ratio (SNR). On the other hand, Alice only shares the CSI for Bob. We also assume that Alice transmits information using 4-QAM through the MIMO channel without frequency selectivity. All channel gains for each transmit symbol are assumed to be independent and constant over four consecutive symbol periods. In the simulations, the aggregated power from all the transmit antennae was normalized to one.

We first compare the secrecy capacity of the proposed scheme with the conventional schemes. It is well known that the secrecy capacity $C_s$ can be expressed by [6]:

$$C_s = \max\left(C_B - C_E, 0\right), \tag{27}$$

where $C_B$ and $C_E$ are the capacities achieved at Bob and Eve, respectively. Because the added AI can be canceled at Bob, $C_B$ can be estimated as in the conventional MIMO systems, as follows:

$$C_B = \log_2 \det\left(\mathbf{I} + \frac{P}{\sigma_b^2} \mathbf{H}\mathbf{H}^{\mathrm{H}}\right), \tag{28}$$

where $P$ and $\sigma_b^2$ are the transmit power and noise variance at Bob, resulting in $P/\sigma_b^2$ being the SNR. On the other hand, $C_E$ is estimated by accounting for the added AI, as follows [18]:

$$C_E = \log_2 \det\left(\mathbf{I} + \frac{P}{(\|\mathbf{W}\|^2 + \sigma_e^2)} \mathbf{G}\mathbf{G}^{H}\right), \tag{29}$$

where, $\sigma_e^2$ is the noise variance at Eve.

We estimate the secrecy performance in terms of $C_s$ in (27), and Figure 3 compares the simulation results of the average secrecy capacity. For comparative purpose, we implement a number of conventional schemes: (1) the Alamouti scheme [15], (2) the LD-QO-STBC scheme [14], (3) the QO-STBC scheme with power scaling [13], (4) the Alamouti scheme with AN [10], and (5) the proposed scheme. Because we assume that Bob and Eve have perfect CSI of their own and the same SNR, the secrecy capacities of the conventional schemes without any AN or AI, i.e., (1), (2), and (3), are almost zero. On the other hand, the addition of AN to the Alamouti scheme contributes largely to enhancing the secrecy capacity, even though Eve has the same SNR as Bob. Moreover, the proposed scheme achieves the best secrecy performance amongst the various schemes.

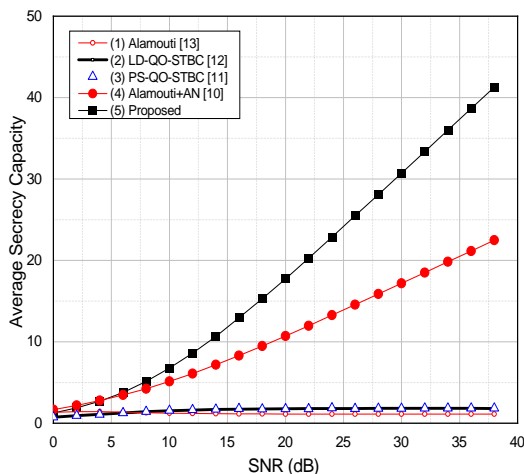

**Figure 3.** Secrecy capacity of the proposed scheme over a Rayleigh fading channel compared to the conventional space–time block coding (STBC)-based PLS schemes.

Next, Figure 4 compares the BER performance simulation results. With the conventional schemes, (1), (2), and (3), Bob and Eve will have almost the same BER performance. On the other hand, due to the highly enhanced secrecy protection in (4) the Alamouti scheme with AN, and (5) the proposed scheme, the BER performance at Eve approaches to 0.5. This means that there is no information leakage to Eve. However, at the expense of power usage of adding AN in scheme (4), we have to sacrifice

power loss. For example, in scheme (4) the same power is allocated to AN as to the information signal, so we can find a 3 dB power loss in the performance. On the other hand, the proposed scheme utilizes the pairwise power reallocation for AI, and thus it does not incur any power loss.

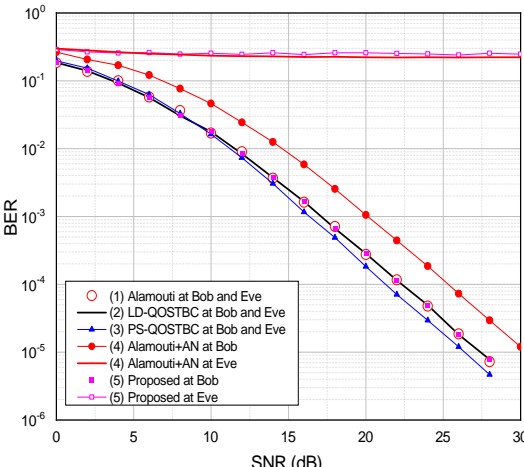

**Figure 4.** Comparison of bit error rate (BER) performances for various STBC-based PLS schemes over a Rayleigh fading channel at the legitimate (Bob) and illegal (Eve) receivers.

## 5. Conclusions

In this paper, we presented a novel secrecy-enhancing scheme by tailoring the waveform of the LD-QO-STBC scheme. The proposed scheme employed dynamic AI that was dependent on the signal, as well as the channel gain of the legitimate receiver. The dependency on the information signal was used to reduce the difference of power allocations across the transmit antennae. In addition, the dependency on the channel gain was used to cancel the AI only at the legitimate receiver, while imposing serious interference on the passive eavesdropper. As a result, the secrecy capacity was highly enhanced. The BER simulation results showed that the proposed scheme prevented information leakage to the illegal listener, even in the case when the illegal listener has a perfect CSI and the same SNR as the legitimate receiver. Because the proposed method does not incur any power loss due to the addition of AI, it can be efficiently utilized for many wireless systems using STBC, resulting in diversity gain as well as security protection. Based on the promising results of the proposed scheme, our future work can be directed to the investigation of dynamic waveform design and efficient detection schemes utilizing artificial intelligence techniques [19,20].

**Author Contributions:** Conceptualization, S.K. and H.L.; methodology, P.S.; software, P.S.; validation, P.S., H.L. and S.K.; writing–original draft preparation, P.S.; writing–review and editing, S.K.; supervision, S.K.; project administration, S.K.; funding acquisition, S.K. All authors have read and agreed to the published version of the manuscript.

**Funding:** The research was supported by Basic Science Research Program through the National Research Foundation of Korea (NRF) funded by the Ministry of Education (2017R1D1A1B03027939).

**Conflicts of Interest:** The authors declare no conflict of interest.

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
