# Peer review of "Waveform Design for Space–Time Coded MIMO Systems with High Secrecy Protection†"

_electronics, doi:10.3390/electronics9122003_

Round 1
Reviewer 1 Report
This Letter claims a Waveform Design for Space-Time Coded MIMO
Systems with High Secrecy Protection. Here are my comments
- The most important thing. Before going towards the technicality of the paper, the author needs to write this paper again. The manuscript has a lot of grammatical and structural mistakes. Some times reviewer feels like it's a translation from some other language into English. For example, check your abstract. The bit error
9 rate performance simulation results presented in this paper reveal that the transmitted information
10 can be properly extracted only at the legitimate receiver. - After Equation 1. Equation 1 seems to be incomplete. ...in a description of Eq 1........where, si denotes.............where is si in Eq.., where is βijs....eq shows some constant while description shows some variable values.
- the proposed methodology, the significance is not reachable. Please explain. how this system is unique and novel
- How channel gains further cancelled the calibrated AI.
- why the fading is not considered in the proposed solution
Author Response
Please refer to the attached letter, where we provide replies to all reviewers.

Reviewer 2 Report
- The article is an extended version of the paper published in ICTC 2020 but I couldn't find the article online. Most of the things are difficult to follow.
- The designed waveform for space-time coded MIMO system should have a specific application
- Why QO-STBC is chosen for the proposed MIMO?
- How BER is calculated for the proposed waveform design?
- References should be improved, cite recent and top articles
Author Response
Please refer to the attached reply letter, and please note that we added all the replies to the reviewers for their easy understanding of our modifications.

Reviewer 3 Report
The authors propose a new waveform design for the MIMO system using space-time block coding (STBC) in order to improve the secrecy and the error rate performance. The paper seems interesting, however I have some suggestions for improving the quality of the work.
- Please, improve Section 2.1. I suggest to make it a longer adding more figures and explanations. You can also add a simple example (maybe numerical), making the section as a brief tutorial. The clarity of this section is crucial for helping the interested reader.
- Please, clarify more the condition in Eq. (17).
- Please, improve the captions of Figures 2 and 3, adding more explanations.
- In order to increase the impact of the work and the number of interested readers, I suggest to discuss that the encoding matrix in Eq. (4) is known as design matrix in several other well-known methods (for regression and classification) and some possible extensions of the Alamouti coding jointly with other machine learning schemes. You can consider the following recent and relevant references for the first and second point respectively,
L. Martino, J. Read, "Joint introduction to Gaussian Processes and Relevance Vector Machines with Connections to Kalman filtering and other Kernel Smoothers", arXiv:2009.09217, 2020.
T. J. O'Shea, T. Erpek, T. C. Clancy, Deep Learning Based MIMO Communications,
arXiv:1707.07980, 2017.
Author Response

(The authors gave the same response as above.)

Round 2
Reviewer 1 Report
N/A
Author Response
Please refer to the attached reply letter.

Reviewer 2 Report
Few comments are considered and updated in the article
Applications of the proposed design should be mentioned and also discuss why this work is essential
Author Response

(The authors gave the same response as above.)
